# A Study on Mineral Oil Hydrocarbons (MOH) Contamination in Pig Diets and Its Transfer to Back Fat and Loin Tissues

**DOI:** 10.3390/ani14101450

**Published:** 2024-05-13

**Authors:** Paula Albendea, Chiara Conchione, Luca Menegoz Ursol, Sabrina Moret

**Affiliations:** 1Analytical Chemistry Laboratory, Gembloux Agro-Bio Tech, University of Liège, 5030 Gembloux, Belgium; 2Department of Agri-Food, Environmental and Animal Sciences, University of Udine, 33100 Udine, Italy; chiara.conchione@uniud.it (C.C.); menegozursol.luca@spes.uniud.it (L.M.U.); sabrina.moret@uniud.it (S.M.)

**Keywords:** mineral oil hydrocarbons, pig diets, back fat, pig loin, bioaccumulation, HPLC-GC-FID

## Abstract

**Simple Summary:**

Mineral oil hydrocarbons are a group of lipid pollutants for which concern has been raised because of different contamination scandals in products intended for human consumption, such as Ukrainian sunflower oil (2007/2008) and chocolate (2018). These contaminants may pose different toxicological risks for humans depending on their structure (e.g., saturated or aromatic) and, therefore, their assessment in food represents a relevant scientific target. This study evaluated mineral oil contamination in different lipid sources used to supplement pig feeds, the final levels of those feeds, and the ability of pigs to accumulate mineral oils in two different tissues that might affect human exposure (i.e., back fat and loin). The results showed that pigs accumulated saturated mineral oils but not aromatic ones and that the deposition was higher in back fat than in loin. The levels of saturated mineral oils in back fat reflected the levels of these contaminants in the corresponding feed, which mainly came from two unknown contamination sources other than the lipid source added. Therefore, identifying and controlling these sources may help prevent high concentrations of saturated mineral oils in pig back fat.

**Abstract:**

This study assessed saturated mineral oil hydrocarbons (MOSH) and aromatic mineral oil hydrocarbons (MOAH) levels in grower–finisher feeds for pigs supplemented with 5% crude palm oil (CP), crude olive pomace oil (COP), olive pomace acid oil (OPA), or a blend of CP and OPA (50:50, *w*/*w*); the contribution of the lipid source to that contamination; and the ability of pigs to accumulate MOH in back fat and loin tissues after 60 days of trial. MOSH and MOAH were analyzed with liquid chromatography (LC)–gas chromatography (GC)–flame ionization detection (FID) after sample preparation. Among the lipid sources, CP had the lowest MOH levels, but CP feeds showed the highest contamination. This, along with the different MOSH profiles, indicated the presence of more significant contamination sources in the feeds than the lipid source. The higher MOH contamination in CP feeds was reflected in the highest MOSH levels in pig back fat, whereas MOAH were not detected in animal tissues. Also, MOSH bioaccumulation in pig tissues was influenced by the carbon chain length. In conclusion, feed manufacturing processes can determine the MOSH contamination present in animal adipose tissues that can be included in human diets.

## 1. Introduction

Mineral oil hydrocarbons (MOH) constitute a complex mixture of liposoluble compounds of petrogenic origin, containing between 10 and about 50 carbon atoms. MOH are considered environmental and processing contaminants, and they can be divided into two groups according to their chemical structure. The first group contains mineral oil-saturated hydrocarbons (MOSH), comprising paraffins (linear and branched alkanes) and naphthenes (alkyl-substituted cyclo-alkanes), whereas the second group contains mineral oil aromatic hydrocarbons (MOAH), which include mono- or polyaromatic compounds with a high alkylation degree [1]. Currently, the information available about MOH toxicity is still controversial, but the evidence suggests that MOSH can accumulate in human organs and tissues, and some MOAH can show carcinogenic, genotoxic, and, mutagenic behavior, particularly polyaromatic species with three to seven rings [2,3,4]. 

Since MOH are non-polar molecules, they exhibit a clear affinity for lipid matrices such as vegetable oils, and, therefore, their occurrence in this type of matrices has been widely studied and reported. Thus, the consumption of vegetable oils can be considered as one of the most important sources of human exposure to these contaminants. Moreover, the use of lipid sources in animal nutrition has been a common practice since the manufacture of the first feeds [5], as they can improve feed efficiency and productive parameters because of their high energetic value [6,7]. Therefore, lipid contaminants present in dietary lipid sources, such as MOH, can be transferred to feeds, leading to animals’ exposure to them. It is well-known that animals tend to absorb and bioaccumulate certain types of liposoluble contaminants because of their high affinity for membrane phospholipids and, even more so, for adipose tissues [8,9]. Thus, the potential presence of MOH in feeds could lead to their accumulation in animal tissues, thereby making meat and meat products a plausible source of human exposure to MOH. Among the different types of meat included in the human diet, the most consumed in the EU since 1990 has been pork and, despite the tendency for its replacement with chicken meat, it is expected to remain in the first position at least until 2030 [10]. However, the information available regarding the presence of MOH in meat in general, and in pork specifically, is limited, with only a few studies conducted to evaluate the transfer from feeds to animals. 

This study evaluated MOH contamination in pig tissues across four scenarios possibly applicable to the commercial production of pork. The scenarios studied differed in the dietary lipid source added to the diets fed during the growing–finishing period of the pigs, and, in one case, in the method of fat addition during feed manufacturing. Therefore, this work evaluated the MOH content in those different lipid sources, in pig feeds, and in two different pig tissues that might be included in human diets. This study could address some of the issued recommendations proposed by EFSA in the last scientific opinion [4], such as a thorough investigation of the sources of the hydrocarbons in food or the evaluation of MOSH accumulation.

## 2. Materials and Methods

### 2.1. Lipid Sources, Pig Feeds, and Animals

In the present study, crude palm oil (CP), crude olive pomace oil (COP), or olive pomace acid oil (OPA) were used as lipid sources in pig feeds intended for the growing–finishing period. The selection of these lipid sources was based on their interest in the swine industry: CP and COP are commonly used in the commercial production of pork, whereas OPA can be an economical alternative to them.

Before the growing–finishing period, more than 100 gilts at 10 weeks of age were fed the same diets for 52 days, firstly with a pre-starter feed with 1.4% animal fat (15 d), and, secondly, with a starter feed with 1.5% CP (37 d). For the growing and finishing periods (60 days in total), the pig feeds were manufactured with 5% of one of the three lipid sources (CP, COP, or OPA) or a blend of CP and OPA (50:50, *w*/*w*; CP-OPA). Therefore, a total of 10 feeds were used including 2 common feeds + 4 grower feeds (CP, COP, OPA, and CP-OPA) + 4 finisher feeds (CP, COP, OPA, and CP-OPA). All the ingredients used for the formulation of the ten feeds are included in the catalog of feed materials approved by The European Commission [11], and the feeds were formulated according to the nutritional requirements of the animals in each growth period. The manufacturing of the feeds was performed only a few days before starting the corresponding feeding period to avoid rancidity problems. The only modification in the manufacturing process among the different feeds was that CP was added through a different method than the rest of the ingredients, whereas COP and OPA were added through the same pathway as the other ingredients. Samples of the three lipid sources and the ten feeds were kept in glass vials with Teflon caps and vacuum packed in different high-barrier multilayer bags, respectively, and kept at −18 °C until their analysis.

Breeding and slaughtering were performed following habitual commercial procedures, always in concordance with the directive 2010/63/EU [12]. The animals had ad libitum access to feed and water during the trial. Throughout the entire growing–finishing period, the pens were fed with CP, COP, OPA, or CP-OPA, even if the type of feed was different (grower or finisher). At the end of the trial, the animals were stunned with CO_2_ and immediately exsanguinated at a commercial slaughterhouse.

### 2.2. Sampling of Pig Tissues

To study the MOH content in pig back fat, a total of 36 female pigs (9 per dietary treatment × 4 dietary treatments) were subjected to sampling. Samples were taken from the dorsal midline between the last rib and the first lumbar vertebrae of each animal. All the back fat samples from the same dietary treatment were mixed and divided to form two replicates with the same weight (always more than 100 g per replicate). Therefore, a total of 8 samples of back fat (2 per dietary treatment × 4 dietary treatments) were analyzed to evaluate MOH levels. Homogenization was performed on each replicate separately. This step consisted of grinding the sample and removing the connective tissue, which remained in a colander after melting the back fat sample in an oven (62 ± 2 °C; 55 min). The melted back fat sample was transferred to glass vials with Teflon caps, which were kept at −18 °C until analysis. 

To evaluate the contamination in pig loin, a total of 32 female pigs (8 per dietary treatment × 4 dietary treatments) were chosen, and one portion of *longissimus dorsi* was taken per animal between the L2 and L5 vertebrae. The samples taken from two different animals were used to conform to each loin replicate, and each replicate contained approximately 500 g of loin. Therefore, a total of 16 samples (4 replicates per dietary treatment × 4 dietary treatments) were used to study the MOH content in pig loin. The samples were ground, vacuum packed in different high-barrier multilayer bags, and kept at −18 °C until analysis.

### 2.3. Determination of MOH in Lipid Sources, Pig Feeds, and Pig Tissues

Regardless of the type of matrix (dietary lipid source, feed, pig back fat, or pig loin), each homogenized sample was prepared and analyzed in duplicate. The mean of the two duplicates was obtained and used as the sample result for the calculations.

#### 2.3.1. Reagents and Standard Solutions

m-Chloroperbenzoic acid (mCPBA), potassium hydroxide, sodium thiosulfate, sodium sulfate, methanol, toluene, *n*-hexane, and dichloromethane (the last two were distilled before use) were provided by Sigma-Aldrich (St. Louis, MO, USA). Ethanol was purchased from Supelco (Bellefonte, PA, USA) and milli-Q water was obtained with a Millipore system (Bedford, MA, USA).

A standard mixture of *n*-C_10–40_ *n*-alkanes (added with *n*-C_50_) in hexane was used to verify GC performance, which contained even-numbered alkanes in the specified range, each at 0.05 mg/mL. An internal standard solution (IS) was used to verify the transfer windows of the fractions from the LC to the GC and to quantify the MOH levels. The IS contained *n*-C_13_ at 0.15 mg/mL, 1,3,5-tri-*tert*-butylbenzene (TBB), *n*-C_11_, cyclohexylcyclohexane (CyCy), *n*-pentylbenzene (5B), 1-methylnaphthalene (1-MN), 2-methylnaphthalene (2-MN) at 0.30 mg/mL, and 5-α-cholestane (Cho) and perylene (Per) at 0.60 mg/mL in toluene. Both standard solutions were obtained from Restek (Bellefonte, PA, USA), and they were stored at −18 °C.

#### 2.3.2. Sample Preparation of Lipid Sources

For the three lipid sources (CP, COP, and OPA), 1 g of a sample was subjected to microwave-assisted saponification (MAS) followed by epoxidation as described by Menegoz Ursol et al. [13]. Briefly, after weighing the sample inside a microwave Teflon vessel, 10 mL of *n*-hexane and 10 μL of IS were added, followed by 10 mL of a 1.5 N methanolic KOH solution. The MAS was performed using a Mars 5 from CEM Corporation (Matthews, NC, USA), equipped with 14 GreenChem Plus Teflon vessels. The MAS was carried out at 120 °C for 20 min, with 5 min of previous pre-heating. When the vessels reached the ambient temperature after the cooling phase, 40 mL of water and 3 mL of methanol were added along the walls, and the vessels were kept at −18 °C for 30 min. Once the vessels were at room temperature, the organic phase was transferred quantitatively to a test tube and concentrated to 4 mL. The concentrated organic phase was subjected to a washing step with a methanol–water solution (2:1; *v*/*v*) [13]. Then, the organic phase was collected again and concentrated to 700 µL, which was used to perform the epoxidation. To epoxidate the sample, a 20% (*m*/*v*) ethanolic solution of mCPBA was used and, after stirring the mixture for 15 min, the reaction was stopped using a 10% (*m*/*v*) aqueous solution of sodium thiosulfate and some ethanol [13]. Finally, approximately 600 µL of the organic phase was transferred into an autosampler vial with some anhydrous sodium sulfate.

#### 2.3.3. Sample Preparation of Pig Feeds

The sample preparation of the 10 feeds [2 for the common feeding period + 4 dietary treatments (i.e., CP, COP, OPA, and CP-OPA) × 2 basal diets (i.e., grower and finisher) for the growing–finishing period] was performed with 5 g of sample and consisted of a MAS followed by epoxidation. For feed analysis, the MAS step was carried out after adding 10 mL of *n*-hexane, 15 μL of IS, and 10 mL of a saturated methanolic KOH solution [14]. The MAS program and the steps applied to obtain the phase separation were the same for all the types of samples. However, as the washing step was not necessary for these samples after phase separation occurred, 5 mL of the organic phase was taken and concentrated directly to 700 µL [14]. To finish the sample preparation, the epoxidation was performed as explained before for the lipid fat sources.

#### 2.3.4. Sample Preparation of Pig Tissues

The MOH levels were determined in the eight back fat samples (2 replicates per dietary treatment × 4 dietary treatments) and in the sixteen pig loin samples (4 replicates per dietary treatment × 4 dietary treatments). The analysis of the back fat samples was performed in 1 g of sample, whereas 5 g was used for the loin samples. Both types of tissues were subjected to MAS (with 1.5 N methanolic KOH solution for back fat and a saturated methanolic KOH solution for loin) and the washing step as described before for the lipid sources. Since MOAH were not detected in these samples, epoxidation was not necessary.

#### 2.3.5. Mineral Oil Hydrocarbons Determination

MOH were determined using an online LC-GC-FID system, specifically, LC-GC 9000 from Brechbühler (Zurich, Switzerland). The instrument comprised a high-performance liquid chromatography (HPLC) Phoenix 9000 and a GC Trace 1310 series by Thermo Fisher Scientific (Waltham, MA, USA) with two channels, which allowed the simultaneous detection of MOSH and MOAH. The HPLC system was equipped with an LC column of 25 cm × 2.1 mm i.d. packed with Lichrospher Si-60, 5 µm particle size (Sepachrom, Milan, Italy), and both GC channels comprised an uncoated/deactivated retention gap of 10 m × 0.53 mm i.d. to exploit the retention gap technique [15]. A Y-interface was used as the connection between LC and GC [16,17] and was accomplished by a switching transfer valve system. The retention gap was connected to a GC column by Mega (Legnano, Milan, Italy) of 10 m × 0.25 mm i.d., coated with a 0.15 μm film of PS-255 (1% vinyl, 99% methyl poly-siloxane) through a steel T-piece, which was linked in turn with a solvent vapor exit (SVE) to remove the solvent evaporating during partially concurrent eluent evaporation [18]. The elution of MOH in LC was performed with a flow of 300 μL/min and the gradient program was as follows: 0 min 100% *n*-hexane; 0.1–0.3 min reached 30% dichloromethane. Six minutes after the injection, the column was backflushed with dichloromethane at 500 μL/min for 9 min and then reconditioned at 700 μL/min with hexane for 6.5 min and at 300 μL/min for 1.5 min. The MOSH and the MOAH fractions were eluted from 2.0 to 3.5 min and from 3.8 to 5.3 min, respectively. To perform the GC separation, hydrogen was used as the carrier at a constant pressure of 60 kPa, and the temperature program was raised from 55 to 350 °C at 20 °C/min. To transfer from LC to GC, the carrier gas pressure was increased to 90 kPa. The FID (with a sampling frequency of 50 Hz) and the SVE were set at 360 and 40 °C, respectively. The SVE was closed with a delay of 1.9 and 2.1 min from its opening (start of LC fraction transfer) for the MOSH and MOAH channels, respectively.

#### 2.3.6. Integration and Calculations

The MOSH area was determined by subtracting all sharp peaks, including endogenous *n*-alkanes, from the whole hump of the largely unresolved peaks. Also, for MOAH, all sharp peaks standing on the top of the hump were removed from the total area by manual integration. The position of the baseline was assessed for both types of MOH by procedural blank runs obtained on the same day.

The quantification was carried out with the internal standard method. For MOSH quantification, CyCy was used, whereas, for MOAH, the average value with 5B, 1-MN, 2-MN, and TBB was obtained. All the results were expressed as concentrations in the animal tissue (back fat or loin). The bioaccumulation of MOSH in pig back fat and loin was evaluated with the bioconcentration factor (BCF), which was calculated as the ratio of the MOSH concentration in the tissue (mean of the different replicates of the same dietary treatment) to that in the feed (mean of the grower and finisher feeds of the dietary treatment).

Based on the sample amount processed and injected into the LC-GC-FID instruments, limits of quantification (LOQ) reached for MOSH and MOAH *C*-fractions ranged from 0.05 mg/kg for the loin to around 0.2 mg/kg for the back fat and lipid sources. 

## 3. Results

### 3.1. MOH in Lipid Sources

The total MOH content found for the three vegetable fats used as dietary lipid sources in this study ranged between 24.9 and 149.2 mg/kg (Table 1). Among them, CP showed much lower concentrations of MOSH and MOAH than COP and OPA, whereas these last two fats had similar levels of both types of MOH. The profile of MOSH was similar in the three lipid sources, where the levels of *n*-C_10–16_ and *n*-C_16–20_ fractions were much lower than the rest of the fractions, with a minimum contribution of the former. The MOSH hump was centered around *n*-C_29_, with *n*-C_25–35_ being the main carbon fraction. Regarding MOAH, residual olefins, which remained after epoxidation in the samples, were removed by proper integration. MOAH distribution followed the one observed for MOSH.

### 3.2. MOH in Pig Feeds

Table 2 shows the MOSH and MOAH results obtained for the feeds used during the common feeding period (pre-starter and starter) of the pigs and during the growing–feeding periods, whose ingredient composition corresponded to the basal diet of that period (grower or finisher) plus a 5% of one of the dietary treatments (CP, COP, OPA, or CP-OPA).

The MOH levels varied among the different feeds. The lowest total MOSH content was found for the pre-starter feed, which was half of the amount found for the starter feed. Regarding the grower–finisher feeds, those with CP showed the highest MOSH content, whereas the lowest amount was found for the CP-OPA grower feed. The latter had the same contamination level as the starter feed but with a different distribution in terms of C-fractions (Figure 1). The lowest levels of MOAH were found in the common feeds and in the ones with CP, whereas feeds with OPA showed the highest contents.

As Figure 1 shows, all the feeds had a different MOSH profile from the one observed for the lipid sources. Unlike all the other feeds, the pre-starter feeds showed a unique relevant hump in the range *n*-C_20–50_ (centered on *n*-C_35_), while all the other feeds presented an additional smaller hump centered on *n*-C_22_. All the feeds showed the presence of clusters of peaks in the range of *n*-C_14_ to *n*-C_22_ that resembled those of polyethylene oligomeric polyolefin hydrocarbons (POH) [19,20]. They could result from the contamination of feed ingredients and feeds with microplastics (e.g., during unpacking, or produced by the grinding and shredding processes and end up in feeds). This is supported by some recent works, which have demonstrated the presence of microplastics in feeds and meat, including pork [21,22,23]. 

Regarding the MOAH profile, *n*-C_25–35_ was also the predominant fraction. Unlike CP, which contains limited amounts of endogenous *n*-alkanes, COP contains relevant *n*-alkanes typically distributed in the range *n*-C_21–33_, with odd terms prevailing on the even ones. 

### 3.3. MOH in Pig Tissues

MOAH were not detected in any samples of back fat or loin tissues. The results obtained for MOSH and the BCF are shown in Table 3. Total MOSH ranged from 10.0 to 29.8 mg/kg in back fat and from 0.5 to 1.0 mg/kg in loin, with noticeable variability among the different dietary treatments. MOSH contamination was between ten and fifty times higher in back fat than in loin. Some variability was also observed among BCF values in back fat and loin from different diets.

In terms of the MOSH profile, the main carbon fraction in back fat and loin tissues was again *n*-C_25–35_. Figure 2 compares the MOSH profile among the different types of samples studied (i.e., back fat, loin, finisher feed, and lipid source) when CP was used in the feeds. It can be observed that the first part of the feed’s hump disappeared in the back fat, while the second part of the feed’s hump was shifted to the left. This second part of the hump showed a maximum around *n*-C_32_ in both the back fat and the loin.

As can be observed in Figure 2c, the loin showed the same maximum as the back fat (around *n*-C_32_), but a higher relative percentage of MOSH in the fraction *n*-C_16–25_. Another interesting observation is that some hydrocarbons present in the feed (specifically the ones that eluted between *n*-C_14_ and *n*-C_22_), identified as possible POH, were transferred to the loin. Particularly, in that carbon range (*n*-C_14–22_), the loin showed a hydrocarbons profile very similar to that of the feed, with even-numbered *n*-alkanes prevailing on odd ones (up to *n*-C_20_), accompanied by *n*-alkenes (highlighted by red arrows in Figure 2c). On the other hand, the presence of these hydrocarbons is in agreement with the data obtained by Tejeda et al. [24], who reported the presence of short-chain *n*-alkanes in subcutaneous fat of Iberian cured hams from pigs fed in different management systems. Their results showed the prevalence of even-numbered terms up to *n*-C_20_, with *n*-C_14_ being the most abundant. In our study, the MOSH profile of the loin was characterized by the presence of numerous sharp peaks, which could correspond to the compounds identified by Petron et al. [25] in intramuscular fat from Iberian dry-cured ham. They identified a total of 35 different hydrocarbons, such as branched, cyclic, and unsaturated hydrocarbons, and correlated their presence in the intramuscular fat to the pig diet used. 

## 4. Discussion

### 4.1. MOH in Lipid Sources

MOH levels in oils can range from non-quantifiable levels to exceeding 1000 mg/kg in certain specific oils such as safflower, nut, sesame, or any cold-pressed oils [2,26]. However, most of the vegetable oils studied in the literature corresponded to edible oils, whereas the information about crude oils, such as the CP and the COP used in this study, or other dietary lipid sources for animal feeding, is limited. Considering the information published by EFSA for feed fats [2], the three lipid sources in this study had much lower MOH values than the common ones (lower bound of 254 mg/kg and upper bound of 259 mg/kg). Among the crude oils used in this study, CP showed lower levels of MOSH and MOAH than COP (Table 1), which is in agreement with the information provided by other authors [16,27,28,29,30]. Regarding COP, high MOH contamination levels found in this type of oil were attributed to inappropriate storage and handling of the olive pomace (e.g., long-time exposure to environmental contamination, exhaust of motor vehicles, and leakage of lubricants from the machines used to move the pomace) and to the fact that a large part of the contamination can remain on the pomace, which is easily extracted with solvent and reconcentrated into the little residual oil extracted from the pomace [31]. 

Despite OPA being a by-product, it showed MOH levels similar to those observed for COP (Table 1). OPA is obtained after the acidification of the soap stocks generated during the neutralization step of the chemical refining of COP [32]. Compared with other by-products from edible oil refining, acid oils usually show a higher unsaponifiable matter content [32], which includes tocopherols, tocotrienols, squalene, or lipid contaminants such as MOH. This, together with the neutralization being the step in which most of the contaminants present in the crude oil are removed [33], leads to the hypothesis that OPA should have shown higher MOH contamination than COP. However, our results contradicted this hypothesis, which could be because the OPA used in this study was probably obtained from the chemical refining of a COP with a different composition than the one used in this study. Another factor influencing the MOH contamination in acid oils is the efficiency of neutralization, as it affects the removal of certain compounds present in the crude oil and, therefore, their accumulation in these by-products. Moreover, other subsequent steps of the refining process might be more efficient at removing MOH from the crude oil than neutralization, such as deodorization, which is the last step of the refining process [30,34]. Therefore, the use of acid oils in feed manufacturing does not always imply an increase in animal exposure to MOH. Contrarily, the use of other types of by-products, such as the ones obtained from the deodorization step of the physical or chemical refining (fatty acid distillates and deodistillates, respectively), might be more problematic.

Despite the differences among the lipid sources in MOH content, all of them showed the typical profile of MOSH described in the literature for vegetable oils [29,30,35,36]. This profile suggests that the contamination in the lipid sources may correspond to mineral oil material coming from gas oils (*n*-C_18–35_) [37] and lubricating in contact with the oils along the whole supply chain [38,39]. Regarding the MOAH percentage in the lipid sources, it ranged between 18% (CP) and 25% (OPA), which is in concordance with previously reported data [2,30]. 

### 4.2. MOH in Pig Feeds

The MOH contamination in animal feeds has been less studied than the one in vegetable oils. The study performed by Bauwens et al. [14] showed MOSH and MOAH concentrations in fish feeds similar to the range observed in this study for pig feeds (Table 2). Even though the MOH contamination was comparable, the composition of the feeds for fish [14] was different from the one of the pig feeds in this study, as animal diets are always formulated to cover the nutritional requirements of the specific species. For example, the fish feeds had a much higher fat content (between 15 and 24%) than the ones used in this study for pigs (5% of added fat), and the type of dietary lipid source was different [14].

Different sources of MOH contamination in feeds have been reported in the literature. These include the previously mentioned addition of by-products from edible oil refining, which tend to accumulate MOH (e.g., distilled fatty acids), environmental contamination from the air (though a small contribution is expected), contamination during ingredient harvesting, and the use of binders for the addition of minor components (e.g., premix of vitamins and minerals). As MOH are lipid-soluble contaminants, the main source of contamination in animal feeds might be expected to be the lipid source used to supplement them. However, the results of this study showed that the fat source can be responsible for only a minor percentage of the MOSH contamination found in feeds. Two different major sources of contamination could be distinguished in the MOSH profile of the grower–finisher feeds (Figure 1). 

The first part of the hump (*n*-C_16–25_) was similar for the different grower and finisher feeds, whereas it was higher in those feeds than in the starter feed. At least part of this atypical contamination might be due to some main ingredients, such as some cereals, which are used in higher amounts in diets fed during the growing–finishing period than in diets for previous growth periods. For instance, Matei and Pop [40] observed that grain corn had the highest MOSH and MOAH concentrations (77.3 and 4.68 mg/kg, respectively) among all the raw materials used to formulate a feed, and the main MOSH carbon fractions corresponded to the first hump observed in Figure 1. These MOSH fractions of a low molecular range have been reported to be present in cereals, specifically in rice, because of migration from the jute bags used for their transport and storage [26]. Neukom et al. [41] suggested that MOSH between *n*-C_16_ and *n*-C_26_ could originate from an atmosphere contaminated by incomplete combustion products of heating oil or diesel. However, the amount of MOSH in that range found in pig feeds (Table 2) was too high to be explained by environmental contamination. Instead, mechanized harvesting operations and/or contact with some lubricants could be the cause of this contamination. The use of binders for fines was ruled out because it was not reported in the information on the composition of the pig feeds used in this study (which remained mostly confidential).

Regarding the second part of the MOSH hump (*n*-C_25–50_) observed in the feeds (Figure 1), it was higher in CP feeds than in the other grower–finisher feeds, resulting in the highest MOH contamination levels in these feeds (Table 2). These highly boiling compounds have been previously found in other products such as an oil called “four cereals” (corn, rice, wheat, and oat) by Wagner et al. [42]. In that study, the authors highlighted the resemblance of the MOSH profile with one of the synthetic oils added to gasoline for two-stroke engines. In the case of the grower and finisher feeds evaluated in the present study, the differences in this second part of the hump found between CP feeds and the rest of the feeds may be only due to the different input of this unknown source of contamination during the production process, as all the ingredients except the lipid source were the same and CP was the lipid source with the lowest MOSH contamination. The contribution of the lipid source (added at a 5% level) to the total feed contamination ranged from 3 to 4% for the CP feeds to around 32% for the COP feeds.

The lack of differences in MOH levels and profiles among the feeds used during the growing–finishing period suggested that the contamination observed is systematic in this production system. Both types of diets (grower and finisher) were manufactured with approximately one month of difference, which implied that different batches of ingredients were used, and a significant amount of feed was produced between them in the same facilities. However, the MOSH contamination in grower and finisher diets with the same lipid source was similar. Moreover, the results published by Bauwens et al. [14] showed similar MOH levels and MOSH profiles in fish feeds to the ones obtained in this study, regardless of the differences in feed composition. This suggests that the sources of MOH contamination observed in this study may be common in the feed industry. More detailed studies, covering all the raw ingredients, their packaging, and the installations, should be performed to identify the potential sources of MOH contamination in animal diets to control them properly. 

### 4.3. MOH in Pig Tissues

MOH can be absorbed mainly in the small intestine of mammals and can be distributed through the portal and/or the lymphatic system, always through passive processes [4,43]. The specific mechanisms involved in the distribution and deposition of MOH through various tissues have not been elucidated yet and may depend on the molecular structure of MOH. However, the fact that MOH can be deposited in different animal tissues has been previously reported [4]. Concretely, the MOH data obtained for pig tissues in this study showed an accumulation of MOSH in the back fat and loin, while MOAH levels were below the LOQ in both tissues. This can be explained because the MOAH absorbed by the pigs might have been metabolized and excreted [44,45,46]. The limited information in the literature regarding MOAH toxicokinetics in mammals shows that the mechanism of biotransformation is highly dependent on the molecular structure of the different compounds comprising MOAH (e.g., the number of aromatic rings or the degree of alkylation) [2,4]. For instance, biotransformation by CYP (P450) enzymes may lead to metabolic activation and the formation of reactive and genotoxic metabolites from MOAH with three to seven aromatic rings and with no or low degree of alkylation. This activation mechanism involves the production of an epoxide group in specific positions of the polycyclic ring system, followed by the formation of other reactive electrophilic metabolites capable of binding to cellular macromolecules. This metabolic route is the most critical one from the toxicological point of view of MOAH. However, the tendency of MOAH to undergo that activation process might be affected (blocked or impaired) by the position of the alkyl chains in the aromatic ring. Depending on the degree of alkylation and the position of the alkyl chains, the predominant metabolic route can be the oxidation of the aromatic ring system or the oxidation of the alkyl chain [2,4]. On the other hand, MOAH can be biotransformed into more hydrosoluble compounds, which facilitate their bioelimination. More studies should be performed to understand the different metabolization routes of MOAH in mammals and how the degree of alkylation and the position of the alkyl chains in the aromatic ring can affect the metabolization process.

Regarding MOSH, the results of this study showed a minimum MOSH bioaccumulation in pig loin in comparison to the one in back fat tissue, which was reflected in the BCF values near zero obtained for the loin (Table 3). This may be linked to the lipidic character of MOSH and the lipid percentage of these two different tissues. However, when expressed in terms of fat content (estimated at 1% for loin and 80% for back fat), MOSH contamination was, on average, 13 times higher in intramuscular loin fat compared with back fat. This demonstrates a highly different impact of MOSH exposure in loin and back fat, likely because of the differing metabolism of MOSH in these tissues.

According to the EFSA’s opinion published in 2012 [2], animal fat was the second greatest contributor to MOSH chronic exposure among all the different food groups for infants, the elderly, and very elderly European people. This highlights the importance of studying the ability of animals to accumulate MOH and evaluating the final MOH levels in animal fat tissues that might be included in the human diet. The results obtained in this study for the back fat tissues of pigs (Table 3) agreed with the occurrence of MOSH in animal fat reported by EFSA [2]. The values obtained for back fat samples coming from the COP, OPA, and CP-OPA dietary treatments (Table 3) were between the 75th percentile (10.0 mg/kg) and the mean (22 mg/kg and 24 mg/kg for lower and upper bounds, respectively) [2]. Among the different dietary treatments, the back fat tissues from pigs fed with CP feeds showed the highest MOSH content (Table 3), reflecting the differences in MOSH contamination between CP feeds and the ones with the other lipid sources (Table 2). Even if the MOSH levels of CP back fat were not much higher than the mean reported by EFSA for animal fat [2], the outcomes of this study suggested that the manufacturing process of animal feeds can have a noticeable impact on MOSH contamination in animal fat tissues. However, the different levels of MOSH contamination in feeds were not reflected in pig loin tissue because of the low MOSH levels observed in it, which could have been more critical for human exposure because of its high consumption. Nonetheless, finding quantifiable amounts of MOSH in animal tissues with such a low-fat content highlights the importance of considering all types of animal food products included in human diets for a proper evaluation of human exposure to these contaminants.

Despite the different MOSH contents, all the back fat samples showed the same profile, which reflected the tendency of pigs to deposit MOSH depending on their carbon chain length. As Figure 2 shows, MOSH with a carbon chain length lower than *n*-C_25_ and higher than *n*-C_40_ observed in the feeds, were not highly deposited in the back fat. Regarding low chain length MOSH, some studies have shown that *n*-C_16_ and *n*-C_18_ alkenes can be oxidized in the intestinal mucosa of different animal species, whereas others have suggested that MOSH below *n*-C_20_ are easily eliminated by exhalation [46,47,48,49]. The low accumulation of MOSH between *n*-C_20_ and *n*-C_25_ in the back fat (Table 3 and Figure 2) might be explained by a higher preference for deposition in pig tissues other than the adipose tissue. For instance, Barp et al. [46] observed different preferential bioaccumulation patterns of MOSH in various human organs and tissues, and that MOSH between *n*-C_20_ and *n*-C_29_ were preferentially retained in human liver tissues. Regarding MOSH with high chain lengths, the information currently available in the literature supports the poor absorption of alkanes and MOSH above *n*-C_35_ in humans [46,47,50]. According to Barp et al. [46], bioaccumulation in human fat involves MOSH from *n*-C_16_ to *n*-C_36_ (centered on *n*-C_23_), while in fat from rats, it involves lighter MOSH, from *n*-C_13_ to *n*-C_31_ (centered on *n*-C_18_) [44]. The results of this study showed a very different bioaccumulation pattern in pig back fat, mainly shifted towards higher molecular weight MOSH (from *n*-C_24_ to *n*-C_36_ and centered on *n*-C_32_). This might be explained by possible differences in the ability to metabolize MOSH among different species. Regarding this topic, different metabolic routes in mammals have been reported depending on the molecular structure of MOSH (i.e., linear, branched, or cyclic). All types of MOSH can be metabolized in the small intestine and/or in the liver to the corresponding fatty alcohols, through ω-oxidation of the alkyl chain, and then to fatty acids [2]. Cycloalkanes can undergo ring oxidation, leading to the correspondent cyclanols, which can be subjected to ω-oxidation, whereas the branched alkanes can undergo ω1-oxidation of the branched chain [2].

According to the results obtained in this study, the consumption of pig loin or back fat with the observed MOSH profile (Table 3) will contribute to the levels of human exposure to these contaminants. Even though there is a knowledge gap about MOSH absorption in humans, as mentioned before, evidence suggests that compounds up to *n*-C_35_ can be absorbed by humans [46,47,50]. This means that, based on the carbon chain length of MOSH, more than 70% of the content of pig back fat was within the range that could be absorbed by humans, while in the case of pig loin, this percentage dropped to 55%. However, it is important to highlight that MOSH with the same carbon chain length can have different molecule structures (i.e., linear, cyclic, or branched), which may affect the absorption effectiveness.

The estimation and comparison of MOH accumulation in animal tissues is hindered by the high number of factors that can affect it, such as the metabolism of the species, the physiological condition of the animal, the existence of other MOSH sources apart from the diet, the dose of exposure, or the MOSH profile of the diet [47]. For instance, the results of this study suggest that the BCF factor might be dose-dependent, as CP back fat showed the highest values for the total of MOSH and for the carbon fraction with the greatest tendency to be accumulated in pig tissues (*n*-C_25–35_); however, more studies should be performed to confirm this behavior.

## 5. Conclusions

Lipid sources entering the formulation of grower and finisher feeds had MOH levels in agreement with the literature data, the highest level being found in COP and the lowest in CP. The contribution of the lipid source (added at 5%) to the MOSH contamination found in grower and finisher feeds was calculated to be around 4% for CP, between 21 and 32% for COP and OPA, and between 14 and 19% for CP-OPA. Since it was not possible to analyze all other ingredients, the contribution of each remains unknown. However, as the ingredients used to formulate the different feeds were the same (they were present in approximately the same quantities), it was possible to conclude that the major contribution to feed contamination, in the case of CP, came from feed processing.

This study also showed that the MOH contamination in the pig back fat was impacted by the scenario applied for the commercial production of pork, specifically by the MOH present in animal diets. Concerning the transfer of MOH from food to animal tissues, pigs were able to accumulate MOSH, but not MOAH, in back fat and loin tissues, showing a much higher tendency to bioconcentrate in back fat, when the data were expressed on a wet weight basis but not when expressed as fat content. In the latter case, the MOSH content was on average 13 times higher in loin fat than in back fat. These new data demonstrate the importance of monitoring the presence of MOSH also in muscle tissue, whose MOSH contamination is not negligible (on average 0.7 mg/kg) and should be considered when calculating human dietary exposure. The different MOSH profile of pig back fat compared with the one observed in feeds revealed a trend in bioaccumulation dependent on the carbon chain length of MOSH, in agreement with what was reported by other authors for humans and rats [44,46]. The most relevant result obtained concerning this subject was the different bioaccumulation patterns observed in pig back fat from humans and rats, which regarded MOSH of a higher carbon number (in the range from *n*-C_24_ to *n*-C_36_, centered on *n*-C_32_).

One of the main conclusions of this study is that the manufacturing process of feeds can be a determinant factor in the MOSH levels present in animal fat tissues. More studies should be performed to evaluate in detail the most important sources of MOH contaminations in the feed industry to allow their control.

## Figures and Tables

**Figure 1 animals-14-01450-f001:**
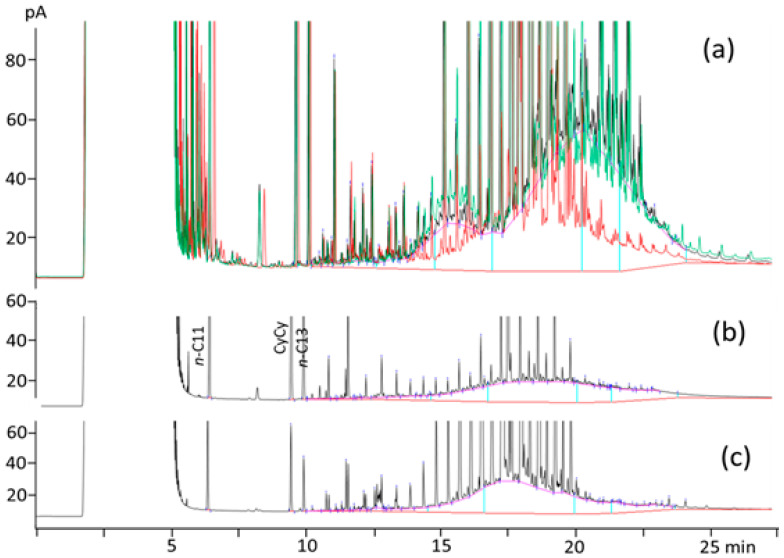
Overlay of MOSH chromatograms obtained for pre-starter feed (red), starter feed (black), and finisher feed manufactured with a blend of crude palm oil and olive pomace acid oil (green) (**a**); MOSH chromatograms of crude palm oil (**b**), and crude olive pomace oil (**c**) used as lipid sources for grower and finisher feeds.

**Figure 2 animals-14-01450-f002:**
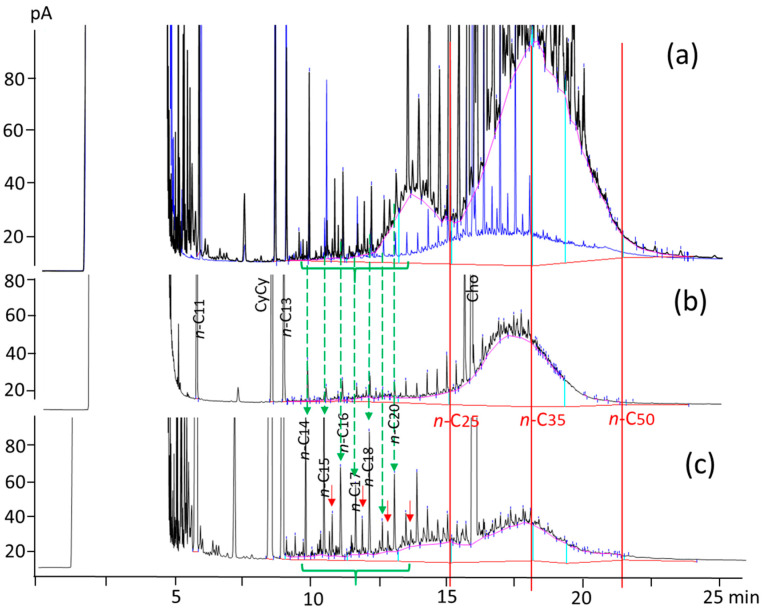
Overlay of MOSH chromatograms obtained for the different types of samples studied when crude palm oil was used in the pig feeds: the lipid source (blue) and the finisher feed (black) (**a**); the back fat (**b**); the loin (**c**). Green arrows highlight the presence of *n*-alkanes, whereas red arrows highlight the presence of *n*-alkenes.

**Table 1 animals-14-01450-t001:** MOSH and MOAH data (mg/kg) of the lipid sources used to supplement the pigs’ grower–finisher feeds.

			CP	COP	OPA
MOSH ^1^ (mg/kg)	*n*-C_10–16_	Mean	0.4	2.0	3.0
		SD	0.01	0.26	0.38
	*n*-C_16–20_	Mean	1.3	6.1	5.4
		SD	0.01	0.04	0.61
	*n*-C_20–25_	Mean	3.6	25.3	21.2
		SD	0.02	1.70	1.13
	*n*-C_25–35_	Mean	11.8	82.8	71.2
		SD	0.24	0.50	1.07
	*n*-C_35–40_	Mean	4.0	17.3	17.0
		SD	0.17	0.65	0.09
	*n*-C_40–50_	Mean	4.2	15.6	10.8
		SD	0.03	1.49	0.62
	*n*-C_10–50_	Mean	24.9	149.2	128.5
		SD	0.43	4.65	2.66
MOAH ^2^ (mg/kg)	*n*-C_10–16_	Mean	<0.2	1.9	1.8
		SD	NC	0.10	0.06
	*n*-C_16–25_	Mean	1.1	16.0	14.4
		SD	0.14	0.29	0.58
	*n*-C_25–35_	Mean	1.9	16.2	15.4
		SD	0.09	0.76	0.89
	*n*-C_35–50_	Mean	2.6	12.3	11.1
		SD	0.30	0.49	0.12
	*n*-C_10–50_	Mean	5.6	46.4	42.7
		SD	0.54	1.54	1.35

Abbreviations: CP, crude palm oil; COP, crude olive pomace oil; OPA, olive pomace acid oil; MOSH, mineral oil saturated hydrocarbons; MOAH, mineral oil aromatic hydrocarbons; NC, not calculated. ^1^ Data are the mean and the standard deviation of two replicate analyses obtained using CyCy as the internal standard. ^2^ Data are the mean and the standard deviation of two replicate analyses, considering in each case the mean of the different internal standards (5B, 1-MN, 2-MN, and TBB).

**Table 2 animals-14-01450-t002:** MOSH and MOAH data (mg/kg) of the pig feeds.

			Common Feeds		Grower Feeds				Finisher Feeds			
			Pre-Starter	Starter	CP	COP	OPA	CP-OPA	CP	COP	OPA	CP-OPA
MOSH ^1^ (mg/kg)	*n*-C_10–16_	mean	0.4	0.3	0.2	0.2	0.3	0.2	0.2	0.3	0.4	0.4
		SD	0.10	0.01	0.12	0.06	0.22	0.03	0.11	0.01	0.06	0.01
	*n*-C_16–20_	mean	0.6	0.9	1.3	1.2	1.4	1.1	1.4	1.3	1.4	1.5
		SD	0.06	0.05	0.04	0.11	0.11	0.04	0.10	0.07	0.05	0.05
	*n*-C_20–25_	mean	1.7	2.5	4.4	4.2	4.5	4.0	4.4	4.4	4.4	4.9
		SD	0.01	0.04	0.36	0.05	0.25	0.23	0.09	0.34	0.48	0.08
	*n*-C_25–35_	mean	5.0	8.3	13.4	10.3	12.3	8.1	15.2	9.3	11.2	11.8
		SD	0.35	0.13	0.90	0.22	0.24	0.11	0.16	1.00	0.60	0.39
	*n*-C_35–40_	mean	1.4	4.7	8.2	4.6	6.0	4.0	9.4	4.5	5.4	5.4
		SD	0.01	0.07	0.70	0.06	0.03	0.15	0.32	0.62	0.63	0.35
	*n*-C_40–50_	mean	1.2	3.6	6.5	3.5	4.8	2.9	7.3	3.3	4.0	4.1
		SD	0.01	0.12	0.56	0.40	0.12	0.22	0.01	0.38	0.18	0.12
	*n*-C_10–50_	mean	10.3	20.3	33.8	24.0	29.3	20.3	37.9	23.1	26.9	28.1
		SD	0.21	0.10	2.37	0.90	0.72	0.29	0.16	2.41	1.95	0.71
MOAH ^2^ (mg/kg)	*n*-C_16–25_	mean	0.2	0.3	0.2	0.6	0.7	0.5	0.3	0.4	0.7	0.6
		SD	0.04	0.01	0.06	0.01	0.11	0.01	0.10	0.04	0.06	0.02
	*n*-C_25–35_	mean	0.5	0.5	0.7	1.1	1.3	1.0	0.6	0.8	1.4	1.0
		SD	0.01	0.02	0.34	0.01	0.09	0.03	0.11	0.08	0.28	0.03
	*n*-C_35–50_	mean	0.4	0.4	0.4	0.6	0.8	0.5	0.4	0.5	0.6	0.7
		SD	0.01	0.01	0.16	0.12	0.20	0.07	0.08	0.02	0.07	0.05
	*n*-C_10–50_	mean	1.2	1.2	1.3	2.3	2.8	2.0	1.3	1.6	2.7	2.3
		SD	0.03	0.01	0.24	0.11	0.01	0.04	0.28	0.14	0.35	0.09

Abbreviations: CP, crude palm oil; COP, crude olive pomace oil; OPA, olive pomace acid oil; MOSH, mineral oil saturated hydrocarbons; MOAH, mineral oil aromatic hydrocarbons. ^1^ Data are the mean and the standard deviation of two replicate analyses obtained using CyCy as the internal standard. ^2^ Data are the mean and the standard deviation of two replicate analyses, considering in each case the mean of the different internal standards (5B, 1-MN, 2-MN, and TBB); MOAH *n*-C_10–16_ was lower than the limit of quantification (0.2 mg/kg) for all the feeds.

**Table 3 animals-14-01450-t003:** MOSH data (mg/kg) and bioconcentration factor in the pig back fat and loin tissues.

			Back Fat				Loin			
			CP	COP	OPA	CP-OPA	CP	COP	OPA	CP-OPA
MOSH ^1^ (mg/kg)	*n*-C_10–16_	Mean	1.0	<0.5	0.6	0.8	<0.05	<0.05	<0.05	<0.05
		SD	0.44	NC	0.35	0.48	NC	NC	NC	NC
	*n*-C_16–20_	Mean	1.4	1.2	0.9	1.2	<0.05	<0.05	0.06	<0.05
		SD	0.14	0.29	0.01	0.25	NC	NC	0.01	NC
	*n*-C_20–25_	Mean	2.2	2.8	1.2	1.3	0.07	0.07	0.15	0.08
		SD	0.07	2.07	0.10	0.10	0.01	0.02	0.06	0.03
	*n*-C_25–35_	Mean	16.6	8.0	4.6	5.6	0.24	0.29	0.37	0.29
		SD	0.07	0.20	1.42	1.63	0.03	0.07	0.15	0.10
	*n*-C_35–40_	Mean	6.2	2.9	1.7	2.1	0.11	0.13	0.2	0.13
		SD	0.38	0.16	0.55	0.73	0.01	0.03	0.03	0.04
	*n*-C_40–50_	Mean	2.4	1.4	1.0	1.2	0.07	0.09	0.22	0.09
		SD	0.40	0.20	0.35	0.50	0.01	0.03	0.18	0.03
	*n*-C_10–50_	Mean	29.8	16.3	10.0	12.2	0.54	0.64	1.02	0.65
		SD	0.19	2.14	2.07	2.23	0.06	0.15	0.44	0.20
BCF ^2^	*n*-C_16–20_		1.1	NC	0.7	0.9	NC	NC	NC	NC
	*n*-C_20–25_		0.5	0.7	0.3	0.3	NC	NC	0.03	NC
	*n*-C_25–35_		1.2	0.8	0.4	0.6	0.02	0.03	0.03	0.04
	*n*-C_35–40_		0.7	0.6	0.3	0.5	0.01	0.03	0.03	0.03
	*n*-C_40–50_		0.4	0.4	0.2	0.3	0.01	0.03	0.05	0.03
	*n*-C_10–50_		0.8	0.7	0.4	0.5	0.02	0.03	0.03	0.03

Abbreviations: CP, crude palm oil; COP, crude olive pomace oil; OPA, olive pomace acid oil; MOSH, mineral oil saturated hydrocarbons; BCF, bioconcentration factor; NC, not calculated. ^1^ Data are the mean, obtained using CyCy as the internal standard, of the different experimental replicates of back fat or loin studied for each dietary treatment (n = 2 for back fat and n = 4 for loin). ^2^ The bioconcentration factor was the ratio between the mean concentration of MOSH in the tissue and that in the feed.

## Data Availability

All data will be made available upon request.

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
