# Peer review of "A Study on Mineral Oil Hydrocarbons (MOH) Contamination in Pig Diets and Its Transfer to Back Fat and Loin Tissues"

_animals, 2024, doi:10.3390/ani14101450_

Round 1
Reviewer 1 Report
Comments and Suggestions for Authors
1. The author should clarify the metabolism of MOH in piglets at different age stages. The author did not specify the age of the experimental animals in the draft, and should study the metabolism of MOH in piglets about to be released.
2.The author should clarify where the substance is mainly absorbed in the gastrointestinal tract.
3.The author should set up a positive control group for this experiment
4.The author should discuss whether feeding this MOH pig will be absorbed by humans
5.The author should discuss more about the mechanism of MOH transfer from the gastrointestinal tract to back fat and loin tissues
6.This study lacks evidence of animal ethics approval.
Comments on the Quality of English LanguageModerate editing of English language required.
Reviewer 2 Report
Comments and Suggestions for Authors
The authors investigated the contamination of fat of pork with MOH. The manuscript is well structured and fairly easy to read, despite deficient English.
The concept of the study was to add tree oils of different MOH concentration to the feeds, two of which are known to have high MOH levels. However, this largely failed, since the feed to which the oils were added was already contaminated at such a high level that the significance of the addition was subject of much uncertainty. Obviously the feed should have been analyzed before adding the oils and replaced by a more suitable one.
For me, the main outcome of the work is the high level of contamination of the feed. There is no description of the composition of this feed, which leaves the question open, where this contamination is from. A simple answer would be that mineral oil was used as a binder for fines, facilitating a homogeneous distribution. In comparison with the MOSH, the MOAH content is low, which would support this. On the other hand, the chromatogram shown in Figure 1 shows a pattern which is atypical for MOH. It should be possible to ask the manufacturer of the feed, what the components were and what kind of an oil was introduced. It is unlikely to be environmental contamination of the feed material.
Another important point is the high level of contamination of the backfat with MOSH. It is not surprising that no MOAH were found, as this agrees with experiments with rats and also human tissues (Barp). The large difference between backfat and loin is another surprise (although somewhere the authors stated that this was expected).
The discussion about the MOH in the oils is less interesting, since this is likely to vary greatly and has no relevant influence on the outcome of this study, since the background contamination was too high.
The authors obviously got into the clinch what to do with the data, as the initial project design probably failed owing to the “wrong” feed used as material to start with. So they expanded on the oils, but this does not reveal much novelty, since the number of samples was too low to enable more generally valuable conclusions.
Some details
45: I would replace 98% by “high proportion” – I’m not sure that this value is well supported.
77: you know how old the guilts were?
78: 15 + 37 days are more than the 37 days mentioned first.
82: the grower feeds and the finisher feeds were the same. Why to distinguish?
82: composition of the CP-OPA?
88: why is this important?
156: was the lipid phase first extracted from the feed?
196: the vapor exit was heated to 360°C? When was it closed?
Table 1: The CP is more contaminated than expected. A comment? The MOH contained a high proportion of MOAH. An explanation? Removal of the n-alkanes in the integration?
Table 2: The starter feed consisted of what? Source of the fat?
244: since CP contained less MOH than the others, why would the MOH content on the feed so high?
Figure 1: The sharp signals of hydrocarbons in the pre-starter feeds are not typical for MOSH. The same seems to hold true for the supplemented starter feeds. An explanation?
Table 3: in the backfat of the animals fed with the CP, the MOSH are highest, obviously not because of the CP. For loin, it is different and with added COP it is highest, as would be expected. This is probably the result of variations in the basic feed and too low additions of oils relative to the background, which caused the experimental design to fail.
286, Fig. 2: not so clearly: I do see the early eluted (not eluting!) ones, but for the later eluted ones, I see a shift, as also in (b).
Fig. 2: the red arrows do not seem to point out n-alkanes
288: the even numbered n-alkanes are probably from PE
297: this section primarily confirms that concentrations vary, and since not many data are provided, this is not an important contribution to science. As a consequence, there is much guessing.
305: [22] These were oils mixed with mineral oil wastes at waste collection sites. It is unlikely to fit here.
313: why don’t you report your own data on COP?
336: exactly: little can be said when there are few data for something that varies in a broad range.
360: Here it would be beneficial to mention that five sources of MOH in feeds are known: environmental contamination from the air, contamination by the harvesting, addition of by-products from oil refining reconcentrating the MOH (e.g. physical refining), addition of oil with admixed mineral oil from waste collection sites (hopefully no longer) and the addition of oil for the addition of fine minor components, like vitamins. However, on the basis of the available information on the feed used, little can be done with it.
Then the paragraph continues with collecting scattering data without sufficient background.
407: systematic only for the given producer
444: “the outcomes of this study suggested that the manufacturing process of animal feeds can have a noticeable impact on the MOSH contamination in animal fat tissues” This is pretty obvious.
452: not surprising when the basic contamination of the feed was the same
It seems to me that not having analyzed the feed before adding the oil was a major mistake. This cannot be corrected. However, I do not understand why the authors did not change the direction of the study and analyze the components this feed was composed of. Obviously they had contacts to the feed producers. I think that this should be done before the manuscript is accepted. Then the main outcome is no longer related to the added oils, but on the feed and its bioaccumulation in the animal – for which there is no data so far.
Comments on the Quality of English Languagewould benefit from improval.
Round 2
Reviewer 1 Report
Comments and Suggestions for Authors
Unable to find author's point-to-point response to comments
Comments on the Quality of English LanguageMinor editing of English language required
Author Response
The authors would like to acknowledge the reviewer for finding time to review our manuscript and for all the suggestions made. The changes made in our manuscript after receiving reviewer’s comments are highlighted in yellow.
- The author should clarify the metabolism of MOH in piglets at different age stages.
The authors agree that there is a gap of knowledge about the metabolism of mineral oil hydrocarbons (MOH) in mammals in the literature and that it is interesting to cover it. One of the challenges in studying MOH lies in their diverse chemical composition, encompassing various compounds with distinct molecular structures. For instance, saturated mineral oils (MOSH) can exhibit linear, branched, or cyclic configurations, while mineral aromatic hydrocarbons (MOAH) can vary in the number of rings and degree of alkylation within both classes. These structural disparities can lead to differences in the metabolism of specific compounds present in the MOH contamination of the pigs’ diets. Given the compositional complexity of MOH, most of the studies in the literature targeted only one type of compounds and they are performed mainly in rodents:
- The limited information regarding the MOSH metabolization in mammals indicates that all MOSH can be metabolized into corresponding fatty alcohols via ω-oxidation of the alkyl chain, a process mediated by the cytochrome P450 system, followed by conversion to fatty acids. This metabolic pathway can occur in both the small intestine and the liver. Furthermore, cycloalkanes may undergo ring oxidation, yielding cyclanols, which are susceptible to subsequent ω-oxidation. Similarly, branched alkanes may undergo ω1-oxidation of the branched chain. Notably, this latter metabolic pathway results in tertiary alcohols, which appear to be a metabolic dead end This last metabolic route drives to tertiary alcohols, which seems to be a metabolic dead end (EFSA, 2012). We have added some comments regarding this information to the discussion of the corrected manuscript (Discussion; L499-506).
- Regarding the metabolism of MOAH, the limited studies on toxicokinetics in mammals have suggested that MOAH are readily absorbed and undergo extensive biotransformation, particularly those with a low degree of substitution, thereby minimizing their accumulation. However, the specific mechanism of biotransformation may vary depending on the molecular structure of the individual compounds within MOAH. For instance, biotransformation mediated by CYP (P450) enzymes may result in metabolic activation and the generation of reactive and potentially genotoxic metabolites from MOAH containing three to seven aromatic rings and little or no degree of alkylation. This bioactivation typically involves the production of epoxides at specific positions within the polycyclic ring system, followed by the formation of other reactive electrophilic metabolites capable of binding to cellular macromolecules. However, this process might be affected (blocked or impaired) by the position of the alkyl chains in the aromatic ring. Depending on the degree of alkylation and on the position of the alkyl chains, the predominant metabolic route can be the oxidation of the aromatic ring system or the oxidation of the alkyl chain (EFSA, 2012, 2023). We have added some comments regarding this information to the discussion of the corrected manuscript (Discussion; L 432-451).
- The author did not specify the age of the experimental animals in the draft, and should study the metabolism of MOH in piglets about to be released.
The authors want to thank you for this comment, the age of the animals has been specified in the manuscript (Material and methods; L 85).
The experimental design of this study aimed to evaluate the MOH contamination in different pig tissues that can be a part of the human diet to investigate whether this contamination originated from the dietary lipid source added to the pigs' diet. To achieve this purpose, the ingredients used to formulate all diets and the conditions employed in the study were typical of those used in commercial pig meat production.
As depicted in Table 2 of the manuscript, the pre-starter diet exhibited the lowest levels of MOH contamination (10 mg/kg), with the shortest duration of exposure (12 days). However, to thoroughly investigate MOH metabolism in piglets, animals would ideally be exposed to higher doses of MOH. While we acknowledge the scientific significance of exploring MOH metabolism in pigs, it is essential to recognize that a specific experimental design tailored for this purpose would likely deviate significantly from the commercial conditions implemented in this study. Principio del formulario
- The author should clarify where the substance is mainly absorbed in the gastrointestinal tract.
We concur on the necessity of delineating the primary site of mineral oil hydrocarbon (MOH) absorption. According to the literature, the small intestine is identified as the principal site of MOH absorption, primarily facilitated by passive processes (Miller et al., 1996). This information has been incorporated into the revised manuscript (Discussion, L425-430).
- The author should set up a positive control group for this experiment
As the primary objective of the study was to evaluate mineral oil hydrocarbon (MOH) contamination in pig tissues from animals raised under typical conditions for commercial pig meat production, the experimental design did not include the use of a highly contaminated diet, which would be inconsistent with the commercial conditions replicated in this study.
- The author should discuss whether feeding this MOH pig will be absorbed by humans
We would like to thank you for this suggestion. While specific studies on the rate of absorption of MOH mixtures or even of individual compounds are lacking, insights from the composition of mineral oil saturated hydrocarbons (MOSH) detected in human abdominal fat and human milk suggest that absorption in humans may extend up to n-C35. We have added a paragraph discussing the possible human absorption (Discussion, L507-516).
- The author should discuss more about the mechanism of MOH transfer from the gastrointestinal tract to back fat and loin tissues
The information on this topic in the literature is very limited. As published in the latest EFSA report, the distribution of absorbed MOH can occur through the portal and/or the lymphatic system, always through passive processes (EFSA, 2023). We have added this in the corrections of the manuscript (Discussion, L425-430).
- This study lacks evidence of animal ethics approval.
The ethical evaluation of this study was conducted by the Ethics Committee on Animal and Human Research of the Universitat Autònoma de Barcelona. They confirmed that ethics approval was not necessary for this study because all procedures were carried out under commercial conditions (including ingredients, farm and housing conditions, slaughter, and processing) and were conducted in compliance with Directive 2010/63/EU (EU, 2010). An official letter from the mentioned committee will be provided for the records of the journal.
- Comments on the Quality of English Language Moderate editing of English language required.
Corrections have been made to improve the English language.
References
EFSA, 2012. Scientific opinion on mineral oil hydrocarbons in food. EFSA Journal. 10, 2704. https://doi.org/10.2903/j.efsa.2012.2704
EFSA, 2023. Update of the risk assessment of mineral oil hydrocarbons in food. EFSA Journal. 21, 8215. https://doi.org/10.2903/j.efsa.2023.8215
EU, 2010. Directive 2010/63/EU of the European Parliament and of the Council of 22 September 2010 on the protection of animals used for scientific purposes. Official Journal of the European Union. 276, 33–79.
Miller, M.J., Lonardo, E.C., Greer, R.D., Bevan, C., Edwards, D.A., Smith, J.H., Freeman, J.J., 1996. Variable responses of species and strains to white mineral oils and paraffin waxes. Regulatory Toxicology and Pharmacology. 23, 55–68. https://doi.org/10.1006/rtph.1996.0009

Reviewer 2 Report
Comments and Suggestions for Authors
The authors did a great effort to improve the manuscript. Now, the manuscript probably reached the best possible and the question becomes a yes or no for publication.
Reading the conclusions shows how little was derived from the data:
“This study showed that the use of a dietary lipid source with a lower MOH content 688 (CP) than others (COP or OPA) does not guarantee a lower contamination in animal feeds, 689 as there might be more relevant sources of MOH contamination in the manufacturing 690 process.”
This just says that the dominant contamination has been missed.
“Regarding the transfer of MOH from feeds to animals’ tissues, pigs were able to 691 accumulate MOSH, but not MOAH, in back fat and loin tissues, showing a 692 much higher bioconcentration tendency in the back fat.”
This difference between back fat and loin might be because the concentrations were determined related to the tissue, rather than the fat content. It is obvious that MOSH are accumulated in fat. I do not see an important finding in this.
“The different MOSH profile of 693 pigs’ back fat compared to the one observed in feeds revealed a trend of bioaccumulation dependent on the carbon chain length of MOSH”
This is not new and the relationship has not been shown quantitatively. Looking at the chromatogram by eye, the effect corresponds to the known one.
“Also, the 695 highest MOH exposure of pigs fed with CP diets was reflected in the MOSH levels of 696 the back fat of the pigs, and it might have led to an increase on the ability of pigs to 697 bioconcentrate MOSH in this type of tissue.”
I’m not sure to understand this. Of course, more MOSH in the feed results in more MOSH in the animal. It is less obvious that the bioconcentration factor should have increased – in rats it was shown to decrease.
“Therefore, the main conclusion of this study 698 is that the manufacturing process of the feeds can be a determinant factor in the MOSH 699 levels present in animal fat tissues. More studies should be performed to evaluate in detail 700 the most important sources of MOH contaminations in the feeds industry to allow their 701 control.”
This does not say much. The source of the contamination of the feed could not be clarified. It is not necessarily the “manufacturing process” that is the source of the contamination, and that the contamination of the feed is reflected by the contamination of the animal tissue is no new finding. It would have been great if the authors had contributed information as requested their the last sentence.
In Fig. 2c, the n-alkanes are still incorrectly labeled. The red arrows no not show n-alkanes.
301: the data refer to the fat tissue (probably not the fat) and the loin tissue. Could you inform about the fat content of the loin to explain the difference between BCF and loin? Is the concentration related to the fat content the same?
Table 3: why is the BCF higher, e.g. for C16-C20 and C25-35 than C20-C25. The values should have a homogeneous performance.
I do not know how ambitious this journal is. There is little literature on the contamination of animal feed and related contamination of animal fat. Merely having mentioned this has some merits. The analytical part is sound, though not new – well described by the same group.
However, as a reader I was disappointed, as it could have been an important contribution. The manuscript has a good start, but then I had the feeling to suffer with the authors that the project went wrong.
Comments on the Quality of English LanguageNot the relevant point here.
Round 3
Reviewer 1 Report
Comments and Suggestions for Authors
The author has made detailed revisions according to the revision suggestions.
Reviewer 2 Report
Comments and Suggestions for Authors
The authors did not give up and tried to get the best out of the data they had produced. Mainly the conclusions were amended. The conclusions are fairly transparent in what can be derived from the data and, reading carefully, it is also clear that much is hypothetic - in several points I would disagree – though without having better data.
In all, having gone so far and considering that also the editor allowed for an extra round, I believe that the manuscript should be published.
Comments on the Quality of English LanguageThe text is understandable.